# Kinin B1 Receptor Antagonism Prevents Acute Kidney Injury to Chronic Kidney Disease Transition in Renal Ischemia-Reperfusion by Increasing the M2 Macrophages Population in C57BL6J Mice

**DOI:** 10.3390/biomedicines11082194

**Published:** 2023-08-04

**Authors:** Gabriel Rufino Estrela, Raisa Brito Santos, Alexandre Budu, Adriano Cleis de Arruda, Jonatan Barrera-Chimal, Ronaldo Carvalho Araújo

**Affiliations:** 1Department of Biophysics, Federal University of São Paulo, São Paulo 04039-032, Brazil; brito.raisa@unifesp.br (R.B.S.); alexandre.budu@unifesp.br (A.B.);; 2Department of Clinical and Experimental Oncology, Hematology and Hematotherapy Discipline, Federal University of São Paulo, São Paulo 04037-002, Brazil; 3Department of Medicine, Nephrology Discipline, Federal University of São Paulo, São Paulo 04039-032, Brazil; 4Maisonneuve-Rosemont Hospital Research Center, Montréal, QC H1T 2M4, Canada; barrera.chimal.jonatan.cemtl@ssss.gouv.qc.ca

**Keywords:** chronic kidney disease, acute kidney disease, kinins, inflammation, experimental disease model

## Abstract

Background: Chronic kidney disease (CKD) is a multifactorial, world public health problem that often develops as a consequence of acute kidney injury (AKI) and inflammation. Strategies are constantly sought to avoid and mitigate the irreversibility of this disease. One of these strategies is to decrease the inflammation features of AKI and, consequently, the transition to CKD. Methods: C57Bl6J mice were anesthetized, and surgery was performed to induce unilateral ischemia/reperfusion as a model of AKI to CKD transition. For acute studies, the animals received the Kinin B1 receptor (B1R) antagonist before the surgery, and for the chronic model, the animals received one additional dose after the surgery. In addition, B1R genetically deficient mice were also challenged with ischemia/reperfusion. Results: The absence and antagonism of B1R improved the kidney function following AKI and prevented CKD transition, as evidenced by the preserved renal function and prevention of fibrosis. The protective effect of B1R antagonism or deficiency was associated with increased levels of macrophage type 2 markers in the kidney. Conclusions: The B1R is pivotal to the evolution of AKI to CKD, and its antagonism shows potential as a therapeutic tool in the prevention of CKD following AKI.

## 1. Introduction

Chronic kidney disease is a multifactorial illness that provokes more than 1 million deaths per year worldwide [1]. Acute kidney injury (AKI) triggers inflammatory mechanisms regulated by the kallikrein–kinin system (KKS), often leading to CKD [2]. The KKS exerts its effects through two distinct receptors, known as B1 and B2. These receptors perform different functions. The kinin B1 receptor (B1R), which is absent or expressed at low levels under physiological conditions, is strongly induced under inflammatory circumstances. The B1R is an important mediator in leukocyte responses, especially in neutrophil migration and apoptosis. The B2 receptor (B2R), on the other hand, is constitutively expressed under physiological conditions, but its expression levels decrease during chronic inflammatory diseases [3,4,5,6].

The B1R is expressed in neutrophils, monocytes, T cells, and B cells, and its activity is increased during chronic kidney inflammation. Consequently, the blockade of B1R may be a potential anti-inflammatory therapeutic strategy [7]. B1R is also associated with the liberation of pro-inflammatory components, such as Il-6, and the increase of fibrosis in CKD [8]. The participation of B1R in tubular renal injury induced by cisplatin administration contributes to the inflammatory process in kidney injury, as there is a decrease in creatinine and urea levels in B1KO mice, as well as inflammatory and apoptotic markers [9]. Additionally, Estrela et al., (2017) showed that the B1R blockade prevented the injury effects and the progression of renal disease in a model of kidney injury induced by cisplatin administrations in mice. However, the mechanisms involved in B1R participation in the inflammatory progression during AKI should be better clarified [10].

Another important inflammation element during CKD are macrophages, divided into two major phenotypes, M1 and M2. The M1 macrophages might be activated by mediators in human acute tubulointerstitial injury, while M2 macrophages might contribute to repair and fibrosis in the interstitial compartments [11]. Proinflammatory M1 macrophages participate in the pathogenic accumulation of extracellular matrix in the renal interstitium after chronic epithelial cell injury and increase the capacity of myeloid cell-derived tumor necrosis factor (TNF) to exacerbate renal fibrogenesis and necroptosis [12]. On the other hand, it was observed that lower plasma creatinine and urea levels, protein excretion, and fibrosis were associated with M2 polarization in kidney ischemia/reperfusion injury, where this migration was modulated through the IL4 stimulation [13].

The polarization of M1 to M2 macrophages in the progression of acute kidney injury (AKI) to chronic kidney disease (CKD) is related to the inflammation resolution process, which highly coordinates the actions of various immune and non-immune cells [14,15,16]. The objective of this study is to investigate the role of B1R in the inflammatory mechanisms and its relationship with M1/M2 polarization during the transition from AKI to CKD.

## 2. Materials and Methods

### 2.1. Animals

Male C57BL/6 mice and B1 receptor knockout (B1KO; B1−/−) mice (C57BL/6 background) weighing 22–27 g and aged 9–12 weeks were used for these experiments. The animals were obtained from the Animal Care Facility of the Federal University of São Paulo (UNIFESP). All animals were housed in individual, standard cages and had free access to water and food. All procedures were previously reviewed and approved by the internal ethical committee of the Federal University of São Paulo in accordance with rules issued by the National Council for Control of Animal Experimentation (CONCEA); the project was approved based on the CEUA 34562604192.2. Experimental Protocol.

The mice were divided into the following groups for each experiment: sham group, ischemia-reperfusion group, and ischemia-reperfusion (IR) group treated with R715 (IR + R715).

### 2.2. Ischemia-Reperfusion

The mice were anesthetized with ketamine (91 mg/kg) and xylazine (9.1 mg/kg) i.p before the surgical procedure. A unilateral flank incision was performed to expose the kidney, and the renal pedicle was dissected. The incision was made only in the left flank. We induced renal ischemia by placing non-traumatic vascular clamps over the dissected left renal pedicles for 30 min. The clamps were then released, and the mice received 0.5 mL of 0.9% NaCl (37 °C). Incisions were closed in two layers, with 5–0 sutures, and reperfusion was allowed. The mice were followed for 6 weeks. Twenty-four hours before euthanasia, mice were subjected to nephrectomy of the right kidney to evidence renal dysfunction. For the nephrectomy, the same procedures listed before were employed, the right kidney was exposed, and then two ligatures (5–0 silk) were placed around the renal vessels, each one with a single knot, the kidney was removed, and incisions were closed as previously described. Sham-treated mice were subjected to the same procedure but without renal pedicle clamping.

### 2.3. R715 Treatment

For acute kidney injury, the animals were treated with 800 µg/kg of R715 48, 24, and 1 h before the surgery. For the model of chronic kidney disease, they were treated 48, 24, 1 h before the surgery and received one administration 24 h after the surgical procedure.

### 2.4. Blood Sampling and Tissue Collection

The mice were anesthetized with ketamine (91 mg/kg) and xylazine (9.1 mg/kg) i.p., and blood was collected via cardiac puncture. For serum, blood samples were allowed to clot for 2 h at room temperature and were then centrifuged for 20 min at 2000× *g*. The samples were then stored at −20 °C. Kidney tissue was collected, and the renal capsule was removed. Transversal cuts were performed, and the kidneys were immediately frozen in nitrogen and then stored at −80 °C.

### 2.5. Renal Function

Serum creatinine and urea levels were used to determine renal function. Samples were analyzed using commercially available colorimetric assay kits (Labtest, Lagoa Santa, Brazil). Urine was collected in metabolic cages over 24 h three days before euthanasia, and the protein concentration was determined with a Sensiprot assay kit (Labtest, Lagoa Santa, Brazil).

### 2.6. Real-Time PCR

Kidney samples were frozen at −80 °C immediately after collection. Total RNA was isolated using TRIzol Reagent (Invitrogen, Carlsbad, CA, USA). cDNA was synthesized using the “High Capacity cDNA Reverse Transcription Kit” (Applied Biosystems, Waltham, MA, USA). Standard curves were plotted to determine the amplification efficiency for each primer pair. Real-time PCR was performed using Eva Green System (Solis Biodyne) using specific primers for TNFa, IL1B, IL6, YM1, IL4R, ARG1; the primers were designed using primer3 web, and their specificity was confirmed using NCBI primer BLAST and then synthesized (Exxtend, Campinas, Brazil); their sequences are listed in Table 1.

### 2.7. Tubular Injury and Renal Fibrosis Analysis

The kidneys were fixed in formaldehyde 10% and then dehydrated and embedded in paraffin. Sections (4 µm) were cut and stained with hematoxylin eosin and Sirius red. At least six subcortical fields were visualized and analyzed for each mouse using a Leica DM4000 microscope at a magnification of 200×. The tubular injury score was determined based on the percentage of tubules showing luminal casts, cell detachment, or dilation and assigned according to the following scale: 0 = 0 to 5%, 1 = 6 to 25%, 2 = 26 to 50%, 3 = 51 to 75%, and 4 > 75%. Histology analysis was performed blind to experimental groups to assess tubule-interstitial fibrosis based on the Sirius red-positive area and assigned according to the following scale: 1 to 25%, 2 = 26 to 50%, 3 = 51 to 75%, and 4 > 75%.

### 2.8. Statistical Analysis

All data are presented as the mean ± SEM. Intergroup differences’ significance was assessed by one-way analysis of variance (ANOVA) with Tukey’s correction for multiple comparisons. The value for statistical significance was established at *p* < 0.05. All statistical analyses were performed using GraphPad Prism 8 v0.2 (GraphPad, La Jolla, CA, USA).

## 3. Results

### 3.1. B1R Absence and B1R Antagonism (R715) Preserves Kidney Function after Acute Kidney Injury

After the acute kidney injury was induced, kidney function parameters were measured 24 h post-surgery in animals that received the B1 receptor antagonist (R715) treatment and in B1KO mice. Animals that did not receive the R715 treatment and WT animals showed higher levels of creatinine (Figure 1A), urea (Figure 1B), and injury score (Figure 1C), evaluated by hematoxylin and eosin staining (Figure 1G–I) and evidenced by epithelial cell detachment and the presence of tubular cast formation. However, animals that received the B1 receptor antagonist and B1KO animals showed the lowest levels of creatinine (Figure 1A,D), urea (Figure 1B,E), and tubular injury score (Figure 1C,F), also evaluated by hematoxylin and eosin staining (Figure 1J–L), indicating the protective role of B1R in acute kidney injury.

### 3.2. B1R Absence and B1R Antagonism Preserve Kidney Function in Chronic Disease Kidney Model

Six weeks after the ischemia-reperfusion surgery, the CKD model showed higher levels of creatinine (Figure 2A), urea (Figure 2B), proteinuria (Figure 2C), and fibrosis score (Figure 2D) compared to the sham group, as evidenced by increased red staining in the interstitial area. Kidney parameters were measured in animals that received the B1R receptor antagonist and compared to animals that did not receive the treatment but underwent surgery. Animals that received the B1R receptor antagonist showed lower levels of creatinine (Figure 2A), urea (Figure 2B), proteinuria (Figure 2C), and fibrosis score (Figure 2D), evaluated by picrosirius red staining (Figure 2I–K), indicating the protective effect of B1R antagonism.

After the ischemia-reperfusion surgery, WT animals showed higher levels of creatinine (Figure 2E), urea (Figure 2F), proteinuria (Figure 2G), and fibrosis score (Figure 2H) compared to WT sham animals. However, B1KO animals showed significantly lower levels of creatinine (Figure 2E), urea (Figure 2F), proteinuria (Figure 2G), and fibrosis score (Figure 2H), evaluated by picrosirius red staining (Figure 2L–N), compared to WT animals, six weeks after the ischemia-reperfusion surgery. These results highlight the important role of B1R in the AKI to CKD transition.

### 3.3. B1R Antagonism Ameliorates Resolution Kidney Inflammatory Process through Macrophage M2 Polarization Increase and Macrophage M1 Decrease

A total of 24 h after ischemia-reperfusion surgery, real-time PCR was performed to identify macrophage M1 and M2 markers. Animals that received the B1R antagonist showed lower levels of M1 macrophage markers, including tumor necrosis factor (TNFa) (Figure 3A), interleukin 1 beta (IL1B) (Figure 3B), and interleukin 6 (IL6) (Figure 3C), compared to those that did not receive the antagonist. These findings indicate a decrease in inflammation. Furthermore, an increase was observed in the M2 macrophage markers, including chitinase-like proteins (YM1) (Figure 3D), interleukin 4 receptor (IL4R) (Figure 3E), and arginase 1 (ARG1) (Figure 3F), in the animals that received the B1R antagonist, suggesting an increase in reparative macrophages and thus potential kidney injury resolution.

## 4. Discussion

In this work, we show the role of the B1 receptor in the transition of acute kidney injury to CKD and the involvement of the polarization of M1 and M2 macrophages. The kallikrein/kinin system plays an important role in the immune system and inflammation, and this is evident in acute kidney injury [9]. The B1R’s role in kidney inflammation is known, but the underlying mechanisms should be better elucidated.

We used an ischaemic-reperfusion model for kidney injury and observed the role of B1R 24 h and 6 weeks after surgery through B1KO animals and an antagonist for B1R. The B1KO animals are known to be protected against nephrotoxic injury and inflammation, presenting lower creatinine and blood urea levels than wild-type animals [9]. B1KO animals treated with B1R antagonist have less fibrosis after UUO-induced renal fibrosis [2]. Furthermore, in this fibrosis model, B1KO animals present lower creatinine, proteinuria, and Il-6 [8]. Animals with blocked B1R also had fewer glomerulonephritis effects, such as lower creatinine and fewer cast formations. It is known that glomerulonephritis can be a predecessor to a tubulointerstitial fibrosis. This is a connection with the B1R and an efficient form of preventing chronic fibrotic kidney disease [2].

In an ischaemic-reperfusion injury, creatinine is higher since the first days of injury and is maintained after weeks of injury. Another characteristic of the injury is higher fibrosis and urea blood levels, indicating a kidney function decrease [17,18]. In this work, we have demonstrated the important role of the B1 receptor in AKI development, as we could observe a lower level of creatinine, urea, and injury scores in B1KO animals as well as in animals that received the treatment with B1 antagonist, suggesting the B1R’s contribution to the injury kidney mechanism and also the kidney-protected role of the B1R antagonist.

Another relevant component in this inflammation machinery are the macrophages; they display two subtypes that perform different functions. The M1 macrophage is involved in inflammation processes, while M2 macrophages play an important role in the resolution process. Usually, in the injured kidney, in animals with a larger M1 presence, high blood urea levels were observed alongside higher kidney fibrosis, evidencing more damage. In accordance with these observations, in this work, we found fewer M1 markers in animals that were administered B1R antagonist. This points towards a role of B1R in inflammation and injury of the kidney [12]. Generally, in CKD, a larger migration of M1 macrophages is observed, accompanied by a compromised M2 macrophage migration [19]. In our CKD model, we found lower levels of M1 macrophage markers in animals treated with B1R antagonists after kidney injury, indicating a correlation between the B1R inflammation role and M1 macrophage migration.

The massive M1 presence in the first 24 h after a kidney injury is followed by Interleukin 6 (IL6), Interleukin 1B (Il1B), and proinflammatory cytokines’ presence [20]. After a kidney injury, arginase 1 (arg1) was increased during the repair phase, pointing towards the adoption of a phenotype akin to alternatively activated macrophages (M2). The M2 macrophage multiplication during cell proliferation in the injured kidney tissue suggests the possibility that M2 macrophages promote the proliferation of renal epithelial cells rather than cell death [21]. In addition to that, Li et al. (2015) observed that macrophage polarization was disturbed in rats under CKD conditions, where there were more M1 macrophages and fewer M2 macrophages during the disease [19].

In the present study, we detected that the B1R receptor antagonist improved the levels of M2 macrophage markers supporting kidney functions’ restoration, since we observed no alterations after an ischemic insult in creatinine, urea, and proteinuria levels and lower fibrosis. In other nephropathy types, we can also observe a pattern behavior of macrophage polarization, as in early diabetic nephropathy more migration of M1 macrophages is detected, as in in vitro experiments under high glucose levels. In addition, there are studies that show an increase of urinary markers of kidney associated with macrophage migration (urinary NAG [22], urinary NGAL [11], and urinary LFABP [23]). As soon as the disease advances, there is a change in the macrophage profile to the M2 type, indicating its role in injury resolution [24]. Another study suggests that M2 abundance is linked to the fibrosis process like part of an inflammation resolution [25]. A similar finding was noted in our study, associated with the B1 receptor antagonism, as we also observed an increase of M2 macrophages, which was seen through the macrophage M2 markers YM1, ARG1, and IL4R.

We could observe a marked presence of M2 macrophages after 24 h of surgery upon treatment with B1R antagonists, indicating a resolution of the inflammation process. In animals treated with the B1R antagonist, a major presence of M2 macrophages and less fibrosis, creatinine, blood urea, and proteinuria were observed, in agreement with the main M2 role in the resolution of kidney injury. Additionally, a study observed that M2 increase improved the proliferation of renal epithelial cells, justifying better kidney function parameters. Therefore, the kidney can perform a better filtration, returning to basal levels [21]. YM1 is very important to the immune system and shows an increase when there is a high macrophage stimulation [26]. YM1 was increased during M2 polarization, leading to more fibrosis and higher collagen deposition, indicating YM1′s role in the chronic kidney process and confirming our findings, as we could observe a better renal function accompanied by a YM1 increase [27].

Furthermore, M2 macrophages are also involved in the recovery process after kidney injury; they are involved with reduced glomerulosclerosis, tubular atrophy, and interstitial expansion [28]. Animals that received M2 cell transplantation after kidney injury performed better and presented low levels of creatinine, protein in the urine, lower levels of tubular damage, interstitial volume, and glomerulosclerosis [29]. Il4 is important as a promoter of M2 polarization, so there is strong evidence of inflammation resolution [13,30,31,32]. In addition, arg1 was expressed in active M2 macrophages [21]. It is involved in injury resolution [33,34] and in the downregulation of chronic inflammation, inhibiting cytokine production, bringing to light kidney healing, since it appears increased in our model [32].

## 5. Conclusions

In conclusion, in this study, we could observe the role of the B1 receptor in acute kidney injury and in the transition to chronic kidney disease. We observed a better performance in kidney function parameters following AKI in B1KO animals and after B1 receptor antagonism. Moreover, we could demonstrate the mechanism by which the blockade of B1 receptor can ameliorate kidney injury via an increase of M2 macrophages, suggesting a treatment or support option for AKI and CKD transition.

## Figures and Tables

**Figure 1 biomedicines-11-02194-f001:**
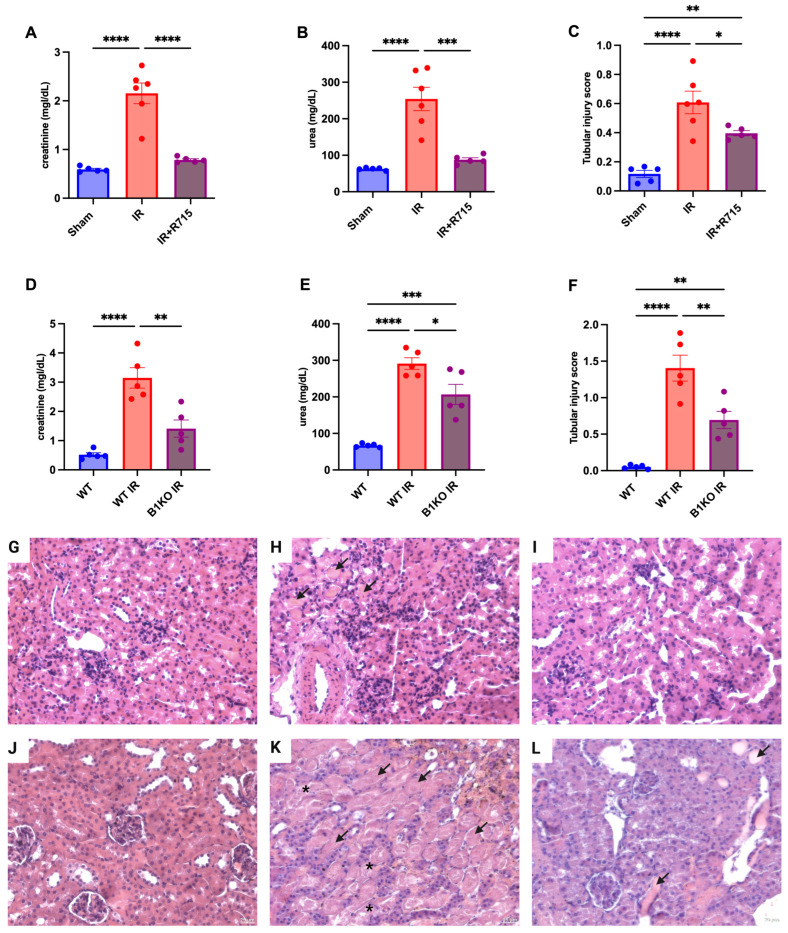
Renal function and tubular injury in acute kidney injury. After 24 h of ischemia reperfusion, renal function was determined by (**A**,**D**) serum creatinine, (**B**,**E**) urea and (**C**,**F**) tubular injury score. Hematoxylin and eosin staining representative images (**G**–**I**) for Sham, IR and IR + R715 and (**J**–**L**) for WT, WT IR and B1KO IR. Data are presented as mean ± SEM; One-way ANOVA followed by post hoc Tukey’s test. Arrows indicates tubular casts, asterisks indicates tubular cell detachment. Scale bars = 25 μm, * *p* < 0.05; ** *p* < 0.01; *** *p* < 0.001; **** *p* < 0.0001.

**Figure 2 biomedicines-11-02194-f002:**
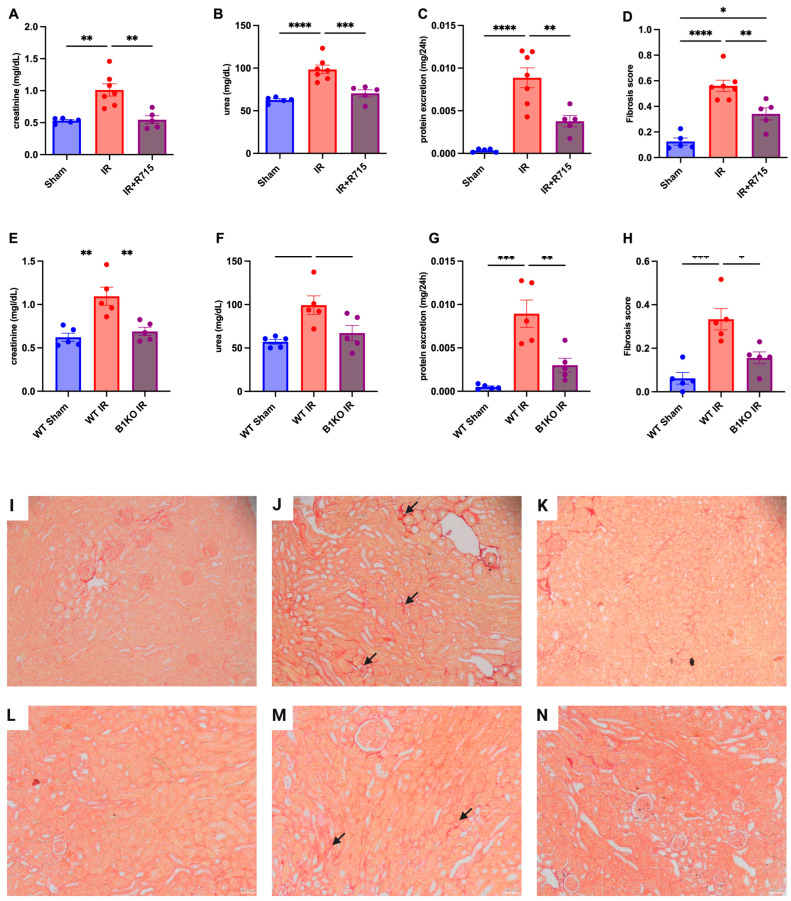
Renal function and fibrosis in chronic kidney disease. Six weeks after IR, renal function was determined by (**A**,**E**) serum creatinine, (**B**,**F**) urea, (**C**,**G**) protein excretion and (**D**,**H**) fibrosis score. Picrosirius staining representative images (**I**–**K**) for Sham, IR and IR + R715 and (**L**–**N**) for WT, WT IR and B1KO IR. Data are presented as mean ± SEM; One-way ANOVA followed by post hoc Tukey’s test. Arrows: interstitial red staining indicating fibrosis. Scale bars = 50 μm, * *p* < 0.05; ** *p* < 0.01; *** *p* < 0.001; **** *p* < 0.0001.

**Figure 3 biomedicines-11-02194-f003:**
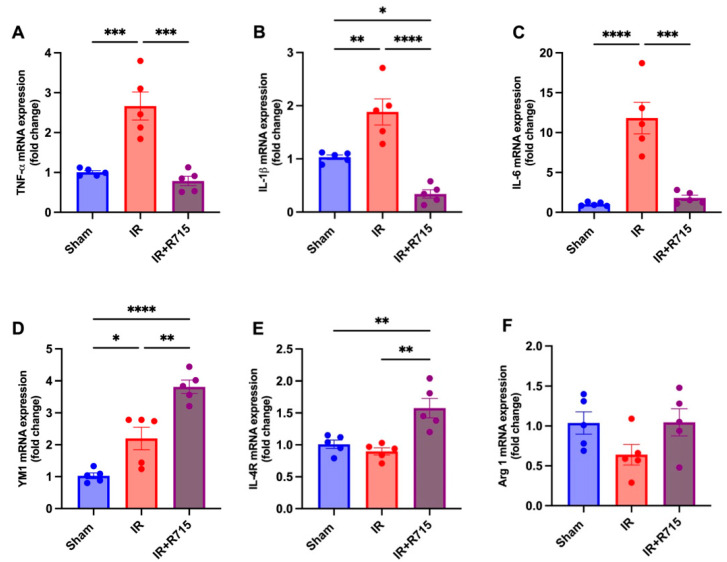
R715 treatment regulates mRNA levels of kidney injury markers and macrophage M2 markers in acute kidney injury. Renal injury was confirmed by increased mRNA levels of (**A**) TNF-α, (**B**) IL-1β, and (**C**) IL-6. (**D**–**F**) Macrophage M2 markers were assessed after ischemia reperfusion surgery (IR). Data are presented as mean SEM; One-way ANOVA followed by post hoc Tukey’s test. * *p* < 0.05; ** *p* < 0.01; *** *p* < 0.001; **** *p* < 0.0001.

**Table 1 biomedicines-11-02194-t001:** Sequences of primers used in real-time PCR assays.

Gene	Forward 5′-3′	Reverse 5′-3′
18S	CGC CGC TAG AGG TGA AAT TC	TCT TGG CAA ATG CTT TCG C
β-actin	CTG GCC TCA CTG TCC ACC TT	CGG ACT CAT CGT ACT CCT GCT T
TNF-α	GCC TCT TCT CAT TCC TGC TTG	CTG ATG AGA GGG AGG CCA TT
IL-1b	AGGAGAACCAAGCAACGACA	CGTTTTTCCATCTTCTTCTTTG
IL-4R	CAC AGT GCA CGA AAG CTG AA	ATG GGC ACA AGC TGT GGT AG
YM1	CCC CTG GAC ATG GAT GAC TT	AGC TCC TCT CAA TAA GGG CC
IL6	TAGTCCTTCCTACCCCAATTTCC	TTGGTCCTTAGCCACTCCTCC
ARG	CGC-CTT-TCT-CAA-AAG-GAC-AG	CCA-GCT-CTT-CAT-TGG-CTT-TC

## Data Availability

All data generated for this study are in this article.

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
