# Peer review of "Kinin B1 Receptor Antagonism Prevents Acute Kidney Injury to Chronic Kidney Disease Transition in Renal Ischemia-Reperfusion by Increasing the M2 Macrophages Population in C57BL6J Mice"

_biomedicines, 2023, doi:10.3390/biomedicines11082194_

Round 1
Reviewer 1 Report
This paper is significant in that it explains the clinical parameters, cytokines and histopathological findings that suggest that a Kinin B1 receptor (B1R) antagonist is useful in the process from AKI to CKD. What should be pursued further is the transition of tubulointerstitial markers of tubulointerstitial damage. Mention of markers such as urinary beta2microglobulin urinary L-FABP urinary NAG urinary NGAL should be added. The description of the histopathological findings is also simplistic and a detailed description of the glomerular, vascular and tubulointerstitial changes should be added.
Reviewer 2 Report
This very interesting and original experimental work demonstrate the role of the kinin/kallicrein system as the trigger of inflammation and lately renal fibrosis in Acute Kidney Injury as the influence of this pathway on macrophage transition (from M2 to M1) and the ability of Kinin B1 receptor antagonist to strongly mitigate this detrimental pathway . My main concern relates to the statistical analysis : although not specified in the text, when analyzing the figures, I guess that each group of experiments contains 5 animals. Thus, its is highly improbable that values of each parameters analyzed are following a Gaussian distribution. Unless authors can prove it using valuable normality tests (Shapiro-Wilk testcoupled with D'agostino-Pearson test), they should give results as median and range and use non-parametric ANOVA (Kruskal-Wallis test) for statistical analyses.
Globally, English wording is good.
